# Role of Ascorbic Acid in the Extraction and Quantification of Potato Polyphenol Oxidase Activity

**DOI:** 10.3390/foods10102486

**Published:** 2021-10-17

**Authors:** Shu Jiang, Michael H. Penner

**Affiliations:** 1College of Food and Pharmaceutical Sciences, Ningbo University, Ningbo 315800, China; 2Department of Food Science and Technology, Oregon State University, Corvallis, OR 97331-6602, USA

**Keywords:** polyphenol oxidase, ascorbic acid, extraction, activity quantification, mechanism-based (suicide) inactivation

## Abstract

The ability to accurately measure the activity of polyphenol oxidase (PPO) in complex matrices is essential. A problem encountered when using spectrophotometric methods is interference due to ascorbic acid (AA), often used as an enzyme “protecting agent” during PPO extraction. This study focuses on the nature of AA’s effect on spectrophotometric determinations of PPO activity as well as enzyme extraction. Potato extracts and semi-purified PPO were used as enzyme sources. The inactivation of PPO attributed to AA is substrate-mediated. The extent of AA-dependent inactivation of PPO in model systems varied between substrates. AA only slows mechanism-based inactivation of PPO induced by catechol, possibly owing to the prevention of quinone formation. AA minimally protects PPO activity during enzyme extraction. The problem associated with AA in PPO assay could be circumvented by using ascorbate oxidase to remove AA when catechol is the primary substrate or by using chlorogenic acid as the primary substrate.

## 1. Introduction

Polyphenol oxidase (PPO) is a copper-containing enzyme widely distributed in plants, fungi, microorganisms, and animals. It catalyzes two types of bimolecular reactions using molecular oxygen as one of the substrates: (1) the hydroxylation of monophenols to *o*-diphenols, sometimes referred to as monophenolase activity, and (2) the oxidation of *o*-diphenols to their corresponding *o*-quinones, sometimes referred to as diphenolase activity. The *o*-quinones resulting from diphenolase activity are highly reactive and often polymerize to form brown/dark pigments known as melanins [1]. Unwanted melanin formation is of widespread interest because it results in the discoloration of plant-based foods, particularly fruits and vegetables [2], which translates into diminished consumer acceptability and ultimately increased food waste. PPO is also important with respect to its natural roles in plant, fungal, bacterial, and animal physiology [3,4] and with respect to its potential biotechnological applications in fields such as food sensors [5], texture development [6], and the treatment of processing waste streams [7].

An underpinning of most PPO-focused research is the ability to accurately quantify the amount of catalytically active PPO in a system. This is particularly important in food sciences where the objective is often to quantitatively compare the PPO activity in different foods and/or assess how PPO activity changes with food processing. Such activity determinations are often done with crude enzyme extracts and/or semi-purified enzyme preparations. The principal methods used for the quantification of PPO activity are spectrophotometric [8], polarographic [9], and chronometric [10]. The spectrophotometric methods are the most common because they are robust, relatively simple, and economical [9]. Such methods are typically based on monitoring the initial accumulation of quinones resulting from PPO-catalyzed oxidation of diphenol substrates; although *o*-quinones are generally unstable, the determination of rates based on their accumulation appears valid when dealing with initial velocity measurements, i.e., where product evolution is negligible [11].

The measurement of PPO activity in a food typically requires that an enzyme-containing extract be prepared. The efficiency of the extraction protocol is important because it determines the extent to which the food’s PPO is available for quantification. A problem associated with extraction, particularly with respect to plant materials, is the comingling of phenolics and PPO that results from tissue disruption and cell decompartmentalization. This comingling can lead to the production/accumulation of reactive quinones, cell constituent modification, and enzyme inactivation [12]. This is important with respect to PPO quantification because extensive PPO inactivation during the preparation and storage of extracts leads to underestimates of the PPO activity in foods. Ascorbic acid (AA) is often added to PPO extracts to minimize quinone accumulation and the related detrimental reactions [9]; quinone accumulation is prevented because AA reduces generated quinones back to their corresponding diphenols [12]. The efficacy of including AA in plant extracts to minimize PPO inactivation during enzyme preparation is not well documented, even though this approach is common [9]. This paper addresses the efficacy of using AA as a PPO protectant in potato extracts.

The presence of AA in PPO extracts can itself be problematic when quantifying PPO activity. In a typical spectrophotometric assay, AA interferes by reducing the generated quinones back to the corresponding *o*-diphenols. This cycle is observed as a lag period prior to quinone accumulation (i.e., prior to a measurable increase in absorbance). The length of the lag phase corresponds to the amount of AA in the system [13]. Once the AA in the reaction mixture is depleted, then quinone accumulation proceeds, absorbance increases, and a reaction rate can be measured. This phenomenon is the basis of chronometric assays used to measure PPO activity [10]. The presence of the lag phase per se is not necessarily bad, but it can lead to underestimates of PPO activity if enzyme inactivation occurs during the lag phase. This is a rational concern based on the widely reported mechanism-based (suicide) inactivation of PPO that occurs with many substrates [14]. If such inactivation occurs, then PPO activities measured following the lag phase will underestimate the PPO activity of the original extract/food. Underestimates of PPO activity can presumably be avoided by removing AA from enzyme preparations prior to activity measurements; dialysis and size-exclusion chromatography are examples of size-based separation methods that have been applied to PPO extracts [15,16]. The least prudent approach to working with AA-containing PPO extracts is to automatically assume that changes in PPO activity occurring during the lag phase are trivial, and can thus be ignored [17]. This paper considers approaches for minimizing lag phase PPO inactivation, and thus improving PPO activity determinations.

Phenol-initiated, mechanism-based inactivation of PPO in the presence of AA is the topic of the previous paragraph. A second type of inactivation that is of potential importance when working with AA-containing PPO extracts is that due to direct PPO–AA interactions. Early experiments with potato PPO suggested that the enzyme is inactivated by AA under anaerobic conditions in the absence of phenolic substrates [18]. A subsequent study concluded there was no direct inactivation of PPO by AA under aerobic conditions in the presence of phenolics [19]. These early studies failed to address the question of whether or not AA inactivates PPO under aerobic conditions in the absence of phenolics. This is relevant because most PPO extraction protocols use some form of phenol adsorbents (e.g., polyvinylpolypyrrolidone, ‘PVPP’), and thus have greatly diminished phenol concentrations [20], yet these same protocols typically result in extracts with oxygen levels at or near air saturation because of the difficulties of working anaerobically. Previous studies questioning the inactivation of PPO by AA under aerobic conditions in the absence of phenolic substrates determined that mushroom PPO is inactivated under such conditions [13,21,22] and pear PPO is not [23]. The limited and contradictory information on this topic prompted us to include in the present study an evaluation of AA-induced PPO inactivation in the presence of molecular oxygen and absence of phenolics.

The focus of this study, as eluded to above, is the influence of AA on the activity of potato PPO under environmental conditions of relevance to the enzyme’s extraction and quantification. Potato was chosen as the enzyme source for the following reasons: (a) potato is an important worldwide food commodity, (b) potato PPO appears to be representative of plant-derived PPOs [24], (c) fresh-cut potatoes brown readily, and (d) previous studies with potato PPO provide insightful information of relevance to the work presented herein [1].

## 2. Materials and Methods

### 2.1. Materials

Russet potatoes (*Solanum tuberosum*) were purchased from local markets. Catechol (*ReagentPlus*^®^, ≥99%), chlorogenic acid (CA) (≥95%), ascorbic acid (AA), ascorbate oxidase (AO), and 2,2′-azino-bis (3-ethylbenzo thiazoline-6-sulfonic acid) diammonium salt (ABTS) were purchased from Sigma-Aldrich (USA). AO stock solution was prepared in aqueous 4 mM sodium phosphate and 0.05% (*w*/*v*) bovine serum albumin, pH 5.6, as described by Bergmeyer [25]. ABTS^+^^●^ stock solution was prepared following the procedure of Huang et al. [26]. PPO substrate stock solutions were prepared in diluted acid (10 mM phosphoric acid) to prevent autoxidation [27].

### 2.2. Preparation of Polyphenol Oxidase Extract

Approximately 50 g, weighed to the nearest 0.1 g, of peeled, washed, diced potatoes (chilled at 4 °C) was mixed in a pre-chilled blender with an equal weight of an aqueous solution (chilled at 4 °C) that was 50 mM sodium phosphate, pH 7.0, and homogenized for 30 s. The resulting homogenate was rapidly filtered through four layers of cheesecloth and Whatman number 1 filter paper via a syringe filter. The filtrate resulting from this process is herein referred to as PPO extract (PPOE). PPOE was diluted with buffer and kept in an ice bath until being assayed for PPO activity.

### 2.3. Preparation of Semi-Purified PPO

A PPO extract was prepared as described just above with the exception that the homogenizing phosphate buffer contained 30 mM AA and the filtrate passing four layers of cheesecloth was centrifuged at 10,000× *g* rpm for 15 min at 4 °C. Ice-cold acetone was added dropwise to the resulting PPO-containing extract until it was 50% acetone (*v*/*v*); precipitation of PPO was allowed to proceed at −20 °C for 1 h. The mixture was then centrifuged at 10,000× *g* rpm for 30 min at 4 °C; the supernatant was discarded and the pellet washed three times with 25 mL ice-cold acetone. The resulting precipitate was dried for 3 h in a hood while sitting on an ice bath. The resulting dry semi-purified PPO powder is hereafter referred to as PPO-acetone powder (PPOA). Prior to enzyme assays, PPOA was added to aqueous 50 mM sodium phosphate (pH 7.0) to give 25 mg PPOA per mL buffer. This suspension was allowed to dissolve for ~1 h at 0 °C and then centrifuged at 10,000× *g* rpm for 15 min (4 °C). The resulting enzyme-containing supernatant was decanted and kept in an ice bath until being assayed; this solution is hereafter referred to as the semi-purified PPO solution.

### 2.4. Quantification of PPO Activity

PPO activity was determined spectrophotometrically using a Shimadzu UV160U spectrophotometer with 1 cm path length cuvettes at ambient temperature (~22 °C) using catechol and CA as substrates. A typical assay system contained substrate (5 mM catechol or 1 mM CA) and enzyme in 50 mM sodium phosphate buffer (pH 7.0). Reactions were initiated by the addition of 0.2 mL enzyme preparation to 2.0 mL temperature-equilibrated substrate solution. The increase in absorbance due to product accumulation was monitored at 400 nm and 420 nm for CA and catechol, respectively. Initial velocities were calculated from the initial linear portion of reaction curves (first ~30 s). PPO activity is defined as change in absorbance per minute (Δabs/min).

### 2.5. Effect of AA on PPO Activity Quantification Using Catechol and CA as Substrates

Enzyme reactions were initiated by adding 0.1 mL of diluted semi-purified PPO preparation to 2.1 mL of temperature-equilibrated substrate solution (final concentration 5 mM catechol or 1 mM CA; 50 mM sodium phosphate, pH 7.0) containing varying concentrations of AA (ranging from 0–0.1 mM). PPO initial velocities were determined as specified above.

### 2.6. Effect of AA on Extents of PPO Mechanism-Based Inactivation in the Presence of Catechol and CA

PPO/catechol incubation solutions were 1 mM catechol, 2 mM AA, and 50 mM sodium phosphate, pH 7.0. Reactions were initiated by adding 20 μL semi-purified PPO solution to 0.2 mL of incubation mixture at 22 ± 1 °C. Reactions were allowed to proceed along with vortexing for defined periods of time before an aliquot of the reaction mixture was removed and tested for residual PPO activity using CA as the substrate (as described above). PPO/CA reactions were initiated by adding 0.1 mL of the aliquot from the PPO/catechol incubation mixture to 2.1 mL of reaction mixture, such that the final solution was 1 mM CA and 50 mM sodium phosphate, pH 7.0. Initial velocities were determined as specified above. Values corresponding to zero-time PPOA/catechol reaction mixture incubation were obtained using reaction mixtures that contained no catechol. Control samples were treated identical to the test samples, except they contained no AA. An analogous set of experiments as described in this section with 0.5 mM CA as the substrate PPO was incubated prior to the measurement of residual activity.

### 2.7. Effect of Using Ascorbate Oxidase (AO) for Ascorbate Oxidation Prior to Quantification of Catecholase Activity

AO solution (0.1 mL, (10 unit/mL) was mixed with 2.1 mL of 5.48 mM catechol, 0.3 mM AA, and 50 mM sodium phosphate, pH 7.0. The resulting mixture was incubated for 10 min at ambient temperature followed by shaking for 30 s to promote oxygen saturation. The PPO reaction was then initiated by adding 0.1 mL five-diluted semi-purified PPO preparation to the AO/AA/catechol solution (final concentration of 5 mM catechol). PPO activity was determined spectrophotometrically as described above. Control experiments were treated identically, but without AO and/or AA.

### 2.8. Direct Effect of AA on PPO Activity in the Model System

These experiments are based on preincubation of PPOA with AA for different periods of time prior to measuring enzyme activity using catechol and CA as substrates. In experiments where catechol was used as the substrate to measure PPO activity, AA was added to five-fold diluted semi-purified PPO preparation to give a final concentration of 1 mM AA. These AA/PPO-containing solutions were incubated for different periods of time (10, 20, 30, 40, 50, and 60 min) at 0 °C. At the specified times, 0.2 mL of buffered AO-containing solution was added to 0.2 mL of the AA/PPO-containing solution (final concentration, 1.25 units AO/mL); these solutions were allowed to incubate for 5 min to remove AA. The resulting solutions were then tested for PPO activity by adding 0.2 mL AO-treated AA/PPO-containing solution to 2 mL substrate-containing solution (final concentration 5 mM catechol). Initial velocities were determined as specified above. Control experiments were treated identically, but without AA.

In experiments where CA was used as the substrate to measure PPO activity, AA was added to semi-purified PPO preparation to give a final concentration of 10 mM AA. AA/PPO-containing solutions were incubated for different periods of time (30, 60, 90, 120, and 150 min) at ~0 °C before being tested for PPO activity. PPO reactions was then initiated by adding 0.2 mL diluted AA/PPO-containing solution to 2 mL substrate-containing solution (final concentration of 1 mM CA). Initial velocities were determined as specified above. Control experiments were treated identically, but without AA.

### 2.9. Direct Effect of AA on PPO Activity in Potato Juice

These experiments are based on preincubation of AA-containing PPOE for different periods of time prior to measuring enzyme activity using CA as substrate. PPOE, 1 mL, was mixed with 0.2 mL AA (final mixture concentration, 4 mM AA) and allowed to incubate for 15 min at ~0 °C, at which time the quantitative PPO assay was initiated by adding 0.2 mL five-fold diluted AA/PPOE mixture to 2 mL substrate-containing solution (final concentration 1 mM CA). Initial velocities were determined as described previously. A zero time-point (i.e., no exposure of PPOE to added AA prior to initiating the reaction) was determined as described above, except the appropriate amount of AA was included simultaneously with the initiation of the PPO reaction. The appropriate amount herein is defined as the estimated amount of AA that would be in the PPO reaction mixture at other time-points. Control experiments were treated identically, but without AA in either the pre-assay incubation or the assay per se. Experiments were done using PPOE preparations from three individual potatoes. Follow-up experiments were done as described above, with the exception that the AA/PPOE mixtures were incubated for longer periods of time (0, 30, 60, 90, 120, and 180 min).

### 2.10. Time Course of AO-Catalyzed AA Reduction in PPOE

AO reactions were initiated by adding 0.3 mL AO solution (final concentration of 3.33 unit/mL) to 1.5 mL five-fold diluted PPOE containing AA (final AA of concentration 2 mM). This reaction was carried out with and without shaking during reaction times at ambient temperature. At selected times, an aliquot of 0.2 mL AO reaction mixture was added to 0.5 mL 0.2 M HCl to terminate AA oxidation prior to the subsequent AA quantification. AA quantification was determined by adding 0.1 mL PPOE/AA solution to 2.0 mL diluted ABTS^+^^●^ solution and immediately measuring absorbance at 734 nm. Control experiments were done identically, with the exception that the PPOE solution contained no AA. The difference between absorbance values for the experimental and control solutions at analogous time points was taken as being directly proportional to AA levels in PPOE (see below). A zero time-point of the AO-treated AA-containing PPOE was determined using the same method, except AO was added following the addition of HCl solutions.

### 2.11. Quantification of AA in the Presence of PPOE Using the ABTS/ABTS^+^^●^ Reaction

A 3.1 mL aliquot of AA-containing 0.2 M HCl (concentrations ranging from 0 to 0.854 mM) was mixed with 1.1 mL diluted PPOE. These solutions, 0.1 mL, were then added to 2 mL of diluted ABTS^+^^●^ solution, followed immediately by measuring the absorbance at 734 nm. Control experiments were done identically, with the exception that the PPOE solution contained no AA. The difference between absorbance values for the experimental and control solutions were taken as being directly proportional to AA levels in PPOE. This is based on immediate reduction of ABTS^+^^●^ by AA compared with other potential reducing compounds [28]. All reactions were done at ambient temperature. The linear calibration curve based on experiments in which differing amounts of AA was added to PPOE solutions ranged from 0 to 0.03 mM AA; the R-value for the calibration curve was 0.9996.

### 2.12. Statistical Analysis

The data were reported as means ± standard deviation of triplicate experiments. Statistical differences between different treatments (groups) were determined by analysis of variance (ANOVA) and Tukey’s post hoc test (*p* ≤ 0.05). Statistical analyses were performed using Minitab 16 (Minitab LLC. State College, PA) software.

## 3. Results and Discussion

Maintaining PPO activity in plant-derived extracts during sample preparation and enzyme quantification is of heightened concern relative to many other enzymes because of PPO’s susceptibility to mechanism-based (suicide) inactivation [14] and the potential for non-specific enzyme inactivation resulting from PPO-catalyzed quinone formation [12]. Figure 1 illustrates the relationship between the two inactivation mechanisms. The typical events associated with the progress of an enzymatic reaction are depicted horizontally. Mechanism-based inactivation is depicted vertically (Figure 1, shown in red) and occurs concomitant with quinone formation [14,29]. Inactivation via reaction of PPO with free quinone is also depicted vertically ((Figure 1, shown in blue); the inactivation per se is generally thought to be a consequence of PPO-quinone adduct formation [30]. The influence of AA on PPO activity loss due to both mechanism-based inactivation and PPO-free quinone interactions is considered herein, first in the context of defined reaction mixtures comprised of semi-purified PPO (PPOA), AA, buffer, and the relevant substrates (catechol and CA), and second in the context of potato enzyme extracts.

Figure 2a shows the relatively rapid inactivation of PPO when catechol serves as the phenolic substrate in the defined PPOA-based system. Catechol was included in this study because it is the most commonly used substrate for PPO quantification [9]. Note that a critical condition required for this experiment is that substrate catechol does not affect the subsequent determination of remaining PPO activity using CA as a substrate (i.e., the amount of catechol from pre-incubation mixture does not compete with CA reactions with PPO for activity measurements). The maximum amount of catechol (~45 μM) present in the PPO/CA reaction mixtures herein did not significantly affect rates of PPO reactions with CA. The data in Figure 2a show that approximately 90% and 75% of enzyme activity is lost after 2 min incubation of PPO with catechol in the absence and presence of AA, respectively. Complete inactivation of PPO occurs with extended reaction times regardless of whether or not AA is present. Activity loss in the absence of AA reflects both mechanisms of PPO inactivation. The loss of activity in the presence of AA reflects that due to mechanism-based inactivation alone, as AA efficiently reduces the product quinones resulting from PPO-catalyzed reactions, thus preventing quinone accumulation and avoiding inactivation due to non-specific PPO-quinone adduct formation. The lack of accumulation of quinones in PPO reaction mixtures containing AA is perceived as an absence of absorbance at wavelengths characteristic of these compounds (see later discussion with respect to ‘lag’ phases in PPO activity measurements).

The data in Figure 2b illustrate inactivation of PPO when CA serves as the phenolic substrate in the presence and absence of AA. CA was included in this study because it is the primary PPO substrate in many plant-based materials/foods, including potatoes [1]. As mentioned above, the same as in the case of catechol as substrate, the maximum amount of CA (~23 μM) present in PPO/CA reaction mixtures did not significantly affect the rates of PPO/CA reactions. The result shows that inclusion of AA in CA-based reaction mixtures provides protection against PPO inactivation, as observed in the catechol-based reaction system (compared with data in Figure 2a). The percent protection due to the addition of AA to CA-based reaction systems is much greater than that observed when AA is added to catechol-based systems. This is a result of CA being a much weaker suicide substrate than catechol. The relative weakness of CA as a suicide substrate is evident from the much faster rates of PPO inactivation in catechol-based versus CA-based reaction mixtures containing AA (recall, PPO inactivation in the presence of AA is solely due to the mechanism-based route). The relative weakness of CA as a suicide substrate means the percentage of total PPO inactivation attributable to mechanism-based inactivation is diminished, thus the percentage of total PPO inactivation attributable to reaction between PPO and free quinone becomes greater. The substrate-dependence of the efficiency of inactivating mechanism-based substrates, as observed here for catechol and CA, is consistent with current understanding (review paper on suicide inactivation). The data in Figure 2 support the premise that the mechanism of suicide inactivation of PPO, regardless of substrate, does not involve reaction with free quinone, otherwise such inactivation would be quenched upon addition of AA to reaction mixtures [14,29].

The combined data in Figure 2 support the conclusion that mechanism-based inactivation is the dominant means of PPO activity loss in systems with catechol as the primary substrate and that reaction between PPO and free quinone is the dominant means of PPO activity loss in systems with CA as the primary substrate.

The preceding paragraphs consider the effect of AA on PPO inactivation in the presence of phenolic substrates. It is also important to consider the influence of AA on PPO activity in the absence of these substrates. This is because PPO extraction protocols often include one or more phenolic sequestrants to extracting buffers [20]. The objective being to lower PPO substrate concentrations, and thus reduce browning and any associated enzyme inactivation. Including sequestrants into AA-containing extraction buffers results in extracts containing PPO and AA, but only minimal amounts of phenolics. The effect of AA on PPOA under these conditions was assessed in this study by pre-incubating the enzyme with different concentrations of AA prior to adding substrate (catechol or CA) to initiate the PPO reaction. Pre-incubation was done at 0 °C to simulate the way an enzyme extract would be treated in an experiment. The data in Appendix A indicate PPO activity (measured as catecholase) is stable in the presence of 1 mM AA and absence of phenolics for up to one hour. A second experiment was done at higher AA concentrations (10 mM), for a longer time (up to 150 min), and using an alternative substrate (CA) to assess PPO activity (Appendix A). The result was the same; there was no evidence of AA inactivation of PPO in the absence of phenolics. No attempt was made to limit molecular oxygen in these experiments (as is the case in typical enzyme extraction protocols). The combined results from these experiments imply that PPO exposure to AA in the absence of phenolics, but in the presence of molecular oxygen, is not detrimental to enzyme activity. Awareness of this result is important because it establishes a difference in the effect of AA on potato PPO activity based on the presence/absence of molecular oxygen; irreversible AA inactivation of potato PPO in the absence of both phenolics and molecular oxygen was established early on [18]. The findings here also illustrate a difference in the susceptibility of potato and mushroom PPO to AA inactivation, as mushroom PPO has been shown to be inactivated by AA in reaction mixtures devoid of phenolics, but containing molecular oxygen [13,21].

A set of experiments was done to test the hypothesis that AA protects PPO from inactivation during enzyme extraction and storage. The data in Appendix A indicate PPO activity in typical potato extracts (PPOE) is equally stable in the short term in both the presence and absence of AA (15 min incubation period; *p* > 0.05). The data in Figure 3a indicate the addition of AA to PPO extracts has, at most, only a nominal stabilizing effect on PPO activity in PPOE preparations stored over longer periods (i.e., measured rates of PPO inactivation with and without supplemental AA were not significantly different over the 180 min test period; *p* > 0.05). The data in Figure 3a also reveal that rates of inactivation under both conditions were not statistically different from zero (*p* > 0.05). These data suggest the protective effect of adding AA to PPO extracts under typical enzyme preparation conditions was not significant over the 3 h test period when working under typical extraction/assay conditions. This result, in light of the protective effects associated with AA addition in Figure 2b, is most simply rationalized by considering the differences in the reaction conditions for the two sets of experiments and the impact of those reaction conditions on the two mechanisms of PPO inactivation (depicted in Figure 1). The rate of PPO inactivation via both mechanisms is dependent on the rate of enzyme catalysis. The rate of enzyme catalysis is largely dependent on the reaction conditions. Reaction mixtures providing the data in Figure 2b, relative to those in Figure 3a, contained more purified PPO preparations, contained higher concentrations of substrate (CA), and were incubated at higher temperatures. The purity of enzyme preparations is important with respect to PPO inactivation resulting from the reaction of the enzyme with free quinone. This is because PPO is in competition with other reaction mixture components for reaction with free quinone. Thus, the more complex the enzyme preparation, the more likely non-PPO components will be present that are amenable to reaction with free quinones; those quinones that react with non-PPO components are not available for reaction with and inactivation of PPO. This competition effectively slows PPO inactivation via any mechanism based on reaction of the enzyme with free quinone. This type of ‘protection’ is the basis of adding extraneous proteins to PPO-containing reaction mixtures to stabilize enzyme activity, as was done in the early days of PPO experimentation [31]. The upshot of this is that compounds endogenous to potatoes, and thus contained in the PPOE preparations, provide protection against PPO inactivation owing to their reaction with free quinone. This minimizes the importance of supplemental AA in PPOE preparations. These same compounds are not present in the semi-purified PPOA preparation.

The second reaction mixture parameter to consider is substrate (CA) concentration. The substrate concentration in the PPOA experiments (which provided the data in Figure 2b) was 0.5 mM CA. CA concentrations in the PPOE preparations were at the very most half this amount. The low CA concentrations in PPOE preparations are dictated by the CA concentration of the raw potatoes, which range from 9.65 to 14.22 mg CA per 100 g fresh weight [32]. Russet potato PPO has a K_m_ for CA of approximately 0.6 mM [33]. Hence, the substrate concentrations in the PPOA- and PPOE-based reaction mixtures are in the range where their differences will impact reaction rates. Rates of PPO-catalyzed CA oxidation are thus higher in the PPOA-based reaction mixtures than in the PPOE-based reaction mixtures. PPO reaction rates and PPO inactivation rates are positively correlated (see Figure 1), thus inactivation rates will also be higher in the PPOA system.

Rates of PPO reactions in the PPOA and PPOE experiments are also expected to differ as a result of differences in temperature. The PPOA reaction mixture was incubated at room temperature (~22 °C); the PPOE preparation was kept on ice to simulate conditions typically employed when working with enzymes. Such a low temperature slows PPO reaction rates as well as PPO inactivation rates in PPOE-based reaction mixtures.

The data discussed to this point show that AA does not directly inactivate potato PPO and that AA minimally protects PPO during its extraction. It is also important to consider the importance of AA during the spectrophotometric quantification of PPO activity. A common problem encountered when spectrophotometrically quantifying PPO activity in enzyme preparation containing AA is that the reaction time course has a lag phase and a subsequent decrease rate of product accumulation (the “initial velocity” following the lag phase) during PPO/catechol reactions, as depicted in Figure 4a. With respect to lag phases, higher concentrations of AA correspond to longer lag phases. Such a lag phase is due to AA reduction of the colored product (*o*-benzoquinone) back to catechol with a consequence of no product accumulated. Product accumulation would not be noticed until AA is exhausted. Thus, AA concentration is positively correlated with the length of the lag phase. The lag phase itself is not a problem during the PPO quantification, but considering the measured PPO activity (i.e., rate of product accumulation), longer lag phases were associated with slower rates of PPO reactions, indicating lag phase-dependent (AA-dependent) PPO inactivation. This definitely underestimates the PPO activity. Similar results have been shown for other PPO/substrate systems [13,34]. However, time courses of PPO/CA reactions with AA (Figure 4c) showed no decreased rates of product accumulation, even following different lag phases. This apparently suggests no AA-dependent PPO inhibition and AA participating in PPO/CA reaction system mainly as a reducing agent (affects reaction product accumulation). The discrepancy in AA effects on PPO reactions with different substrates is plausibly explained by the property of substrates.

Substrate catechol has been shown to be a well-established mechanism-based (suicide) inhibitor of PPOs from several non-potato sources [14,35]. Data presented earlier are consistent with catechol also being a mechanism-based inhibitor of potato PPO (Figure 2a). The mechanism of catechol inactivation of potato PPO does not require product accumulation (Figure 2a). The nature of mechanism-based inhibition is such that a fraction of the PPO-catalyzed catechol oxidations results in enzyme inactivation (see Figure 1). Lag-phase mechanism-based inactivation of PPO is evidenced by a reduction in the rate of product accumulation following the lag phase of the assay (recall the termination of the lag phase corresponds to exhaustion of the AA in the reaction mixture). Such a reduction in the rate of product accumulation is not due to substrate depletion (i.e., O_2_ exhaustion), because adding more O_2_ to the reaction mixture did not increase the rate of product accumulation (Figure 4b). With respect to the PPO/CA-based reaction system, this quantification system did not show significant lag-phase mechanism-based PPO inactivation, most likely because CA is ineffective as a mechanism-based inhibitor of potato PPO compared with catechol (See Figure 2a,b). Nonetheless, PPO inactivation/PPO activity underestimation in the presence of AA is found to be substrate-dependent. Thus, the effect of AA on the spectrophotometric quantification of PPO activity is highly dependent on substrates used in measuring systems, with catechol being more problematic than CA in such PPO activity quantifications.

Quantification of PPO activity in the presence of AA is common as AA is used for preventing color formation/browning in enzyme extracts. The data in Figure 3b illustrate that AA addition does slow the browning of PPOE preparations, as expected based on its use as an anti-browning agent. The result shows that including 30 mM AA during potato PPO extraction prevented browning for at least 2 h; color formed rapidly in potato extracts, while AA was absent. However, when consider PPO activity in potato extracts, the color formation minimally affects PPO activity and enzyme activity remains relatively stable for up to 3 h (see Figure 3a). This is different in potato system as a positive correlation between quinone/browning formation and PPO inhibition has been reported [36]. Thus, during PPO extractions in potato, AA is not necessary to be used to prevent enzyme activity loss (only minimal activity loss was found in potato extraction in the absence of AA), but recommended to be added for maintaining the property of native enzyme by minimizing any significant modification that could be due to *o*-quinone formation/browning reactions [37,38].

Based on the need of using AA during PPO extraction, it is important to consider appropriate approaches for better quantifying PPO activity in an AA-containing system, but still avoid problems with PPO inactivation. As discussed above, the problem associated with AA commonly occurs during a PPO/suicide substrate (e.g., catechol)-based assay (see Figure 4a). In the case of using catechol as a substrate, it is necessary to eliminate AA to retrieve a better activity measurement. In this study, ascorbate oxidase (AO) is introduced to be used for selectively removing AA as it catalyzes the oxidation of AA. Thus, the present study evaluated the effect of using AO for ascorbate oxidation prior to quantification of catecholase activity. Figure 5a shows that measured PPO activity with AA present was 87.6% lower than that without AA. However, after AA was completely removed by AO (no lag phase observed), measured PPO activity was comparable to that obtained from reaction mixture containing no AA. This indicates that AO successfully eliminates AA without affecting PPO activity measurements. In practice, pre-incubating AO with enzyme preparation is normally required when PPO activity quantification is conducted in a crude enzyme extract containing AA. The efficiency of AO-catalyzed oxidation of AA in PPO extract was then evaluated via a time course of AA degradation catalyzed by AO. In this experiment, the amount of AA remaining in AO reaction mixtures at different reaction times was evaluated using AA/ABTS^+^^●^ reaction, during which ABTS^+^^●^ is immediately reduced back to ABTS by AA, so the AA concentration is proportional to the amount of ABTS^+^^●^ reduced. Time courses in Figure 5b showed that AA (2 mM) was completely removed within 5 min by AO with agitation of AO reaction mixtures; while without agitation (i.e., O_2_ supplementation), AO-catalyzed AA oxidation stopped in about 1 min, leading to a high amount of AA remaining in PPOE. This indicates that the amount of O_2_ would be a concern when AO is used for AA removal in PPOE; adequate O_2_ is required to allow continuous AA oxidation catalyzed by AO. However, the amount of O_2_ would not be a concern for subsequent PPO reactions. This is because PPO reaction is initiated by adding only a small amount of PPOE containing AO to a relatively large amount of air-saturated substrate solution. The efficiency of AO-catalyzed oxidation of AA is also associated with other experimental conditions, including AO and AA concentrations; these can be further studied to better serve for PPO activity quantification in an AA-containing system.

Instead of removing AA from a quantification system, another alternative approach to the spectrophotometric quantification of PPO activity in the presence of AA is using CA as an alternative substrate. Recall that PPO/CA-based assay was not sensitive to the reaction mixture containing AA (no significant effect of AA on measured PPO activity on CA after lags, see Figure 4c). Thus, compared with catechol, CA appears to be a more appropriate substrate when AA is present during activity quantification. Figure 6 showed the effects of the amount of PPOE preparation (i.e., enzyme concentration) on PPO activity measurements using CA as substrate. The result showed there was no significant difference in measured PPO activity in PPOE with or without AA present. This suggests no inhibitory effect of AA on PPO/CA-based assay, indicating that CA is a better assay substrate when quantifying PPO activity in enzyme extracts containing AA. Note the fact that PPO activity is commonly reported in catecholase units [39,40], undoubtedly because of the historical use of catechol as a model PPO substrate; thus, for comparative purposes, it may be informative to use catechol as the assay substrate in some cases.

## 4. Conclusions

The aim of the present study was to evaluate the effect of AA on potato PPO activity under environmental conditions relevant to the enzyme’s extraction and activity quantification, thus mainly focusing on the influence of AA on PPO activity loss in the absence and presence of phenolics. The results summarized herein show that AA minimally protects potato PPO activity during enzyme extraction (PPOE-based reaction system), primarily because the enzyme is stable in the absence of AA; AA significantly protects PPO inactivation in a semi-purified system (PPOA-based reaction system), and such a protecting effect is substrate-dependent. No evidence shows AA inactivation of PPO in the absence of phenolic substrates. With respect to the quantification of PPO activity enzyme extracts containing AA, CA is preferred over catechol as the assay substrate owing to the avoidance of mechanism-based inactivation. If using catechol as the assay substrate for such PPO quantification, then it is prudent to use AO to remove ascorbate immediately prior to adding substrate. Thus, the study clearly determines the effect of AA on PPO activity during extraction and quantification, and provides alternative spectrophotometric approaches to avoid underestimates of PPO activity owing to AA.

## Figures and Tables

**Figure 1 foods-10-02486-f001:**
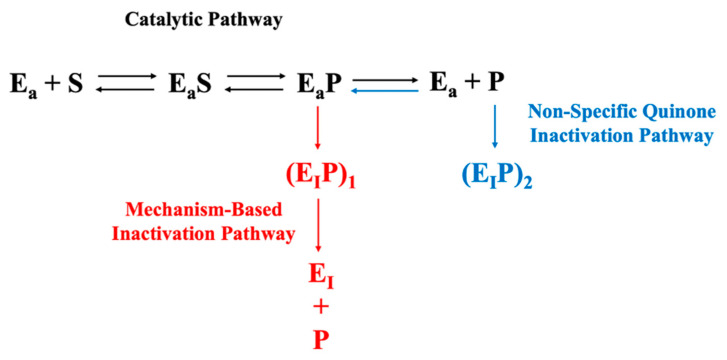
Mechanisms proposed to explain the inactivation of PPO. E_a_, active enzyme; S, suicide substrate; P, product (quinone); E_a_S, active enzyme-substrate complex; E_a_P, active enzyme-product complex; (E_I_P)_1_ and (E_I_P)_2_, inactive enzyme-product complexes 1 and 2, respectively; E_I_, inactive enzyme.

**Figure 2 foods-10-02486-f002:**
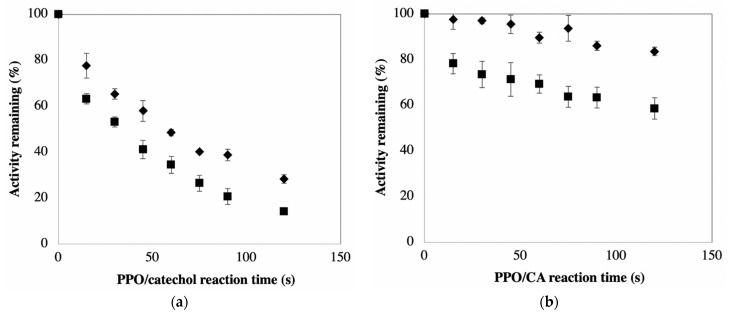
Effect of ascorbic acid (AA) on mechanism-based inactivation of PPO by catechol or CA. Semi-purified PPO was pre-incubated with 1 mM catechol (**a**) or 0.5 mM CA (**b**) in the absence (■) and presence (◆) of AA (2 mM for catechol and 4 mM for CA), in 50 mM sodium phosphate buffer, pH 7.0, for different periods of time. During pre-incubation time, vortexing of reaction mixtures with low speed was included to avoid O_2_ limiting. The remaining PPO activity in the pre-incubation mixture was then measured by adding 0.1 mL pre-incubation mixture to 2.1 mL buffered CA solutions (final concentration 1 mM CA). Values are means ± standard deviations from triplicate assays.

**Figure 3 foods-10-02486-f003:**
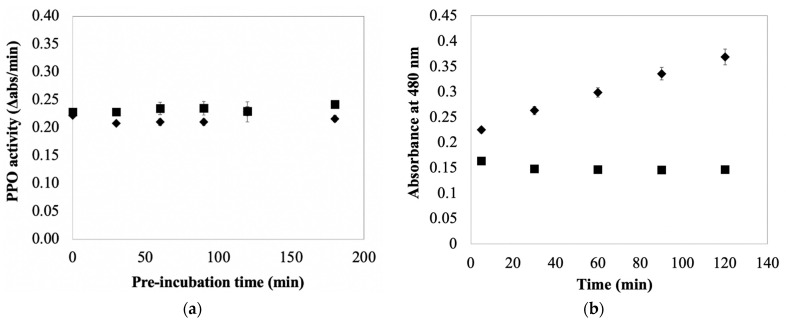
(**a**) Effect of pre-incubating AA with PPOE on subsequent PPO activity measurements using CA as a substrate. PPOE was pre-incubated with (■) and without (◆) AA (AA at 4 mM when present) in 50 mM sodium phosphate buffer, pH 7.0, for different periods of time prior to initiating activity measurements by mixing 0.2 mL pre-incubation mixture (containing PPOE and AA) with 2 mL buffered CA solutions (final concentration 1 mM CA). Control experiments were treated identically, but without AA. (**b**) Time course of color development in potato extract, depicted as an increase in absorbance, in the absence (◆) and presence (■) of AA. Potato extracts were made in 50 mM sodium phosphate buffer containing 0 or 30 mM AA. Color was measured at 480 nm after incubation of potato extracts at room temperature for different periods of time. Values are means ± standard deviations from triplicate assays. Data points without visible error bars have standard deviations smaller than the designated symbols.

**Figure 4 foods-10-02486-f004:**
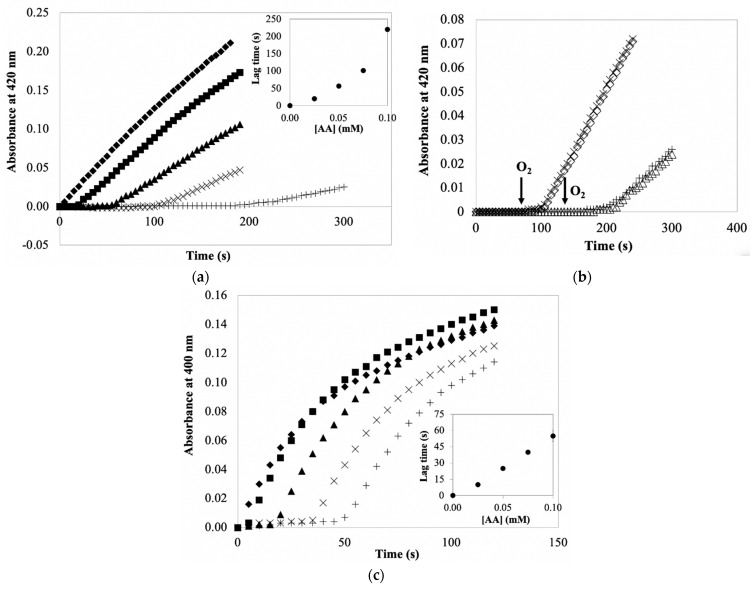
Representative time courses of product accumulation, depicted as an increase in absorbance, during activity measurements of potato PPO using catechol (**a**,**b**) and CA (**c**) as substrates. PPO reaction conditions were 5 mM catechol or 1 mM CA, 50 mM sodium phosphate buffer, pH 7.0, varying concentrations of AA (◆, 0 mM; ■, 0.025 mM; ▲, 0.05 mM; ×, 0.075 mM; +, 0.1 mM). Reactions were initiated by adding 0.1 mL semi-purified PPO preparation to 2.1 mL reaction mixture. (**b**) Reaction conditions were as in (**a**), with the exception of adding O_2_ by 10 s shaking PPO reaction mixtures containing AA (◇, 0.075 mM and △, 0.1 mM) at 1 min or 2 min, respectively, after initiation of the enzyme reaction. Insets in (**a**,**c**) show the relationship between AA concentration and the length of the lag phase.

**Figure 5 foods-10-02486-f005:**
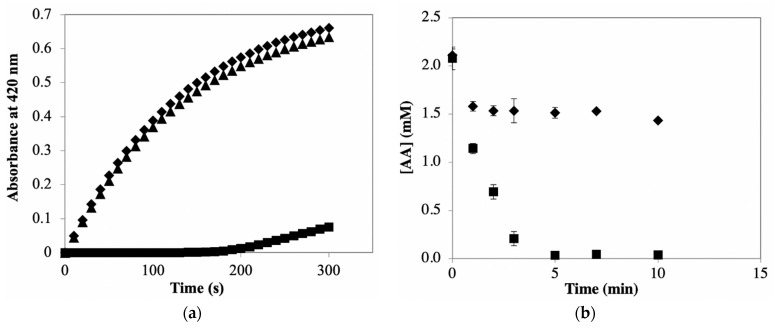
(**a**) Representative time course of product accumulation, depicted as an increase in absorbance, during potato PPO-catalyzed reactions using catechol as a substrate. PPO reaction mixtures were 5 mM catechol solution, in 50 mM sodium phosphate buffer, pH 7.0, containing 0 mM AA (◆), 0.3 mM AA (■), or 0.3 mM AA, and 0.45 unit/mL AO (▲), respectively. Reactions were initiated by adding 0.1 mL semi-purified PPO preparation to 2.2 mL reaction mixture described above. (**b**) Time courses of AO-catalyzed oxidation of AA in PPOE. AO reaction mixtures contained 2 mM AA and five-fold diluted PPOE. Reactions were initiated by adding 0.3 mL AO solution (final concentration of 3.33 unit/mL) to 1.5 mL reaction mixture. AO reactions were carried out with (■) and without (◆) shaking. At specified time, AO reactions were terminated by taking an aliquot of 0.2 mL reaction mixture and adding it to 0.5 mL 0.2 mM HCl solution prior to AA quantification. AA was quantified by ABTS^+^^●^ reduction assay. Values are means ± standard deviations from triplicate assays. Data points without visible error bars have standard deviations smaller than the designated symbols.

**Figure 6 foods-10-02486-f006:**
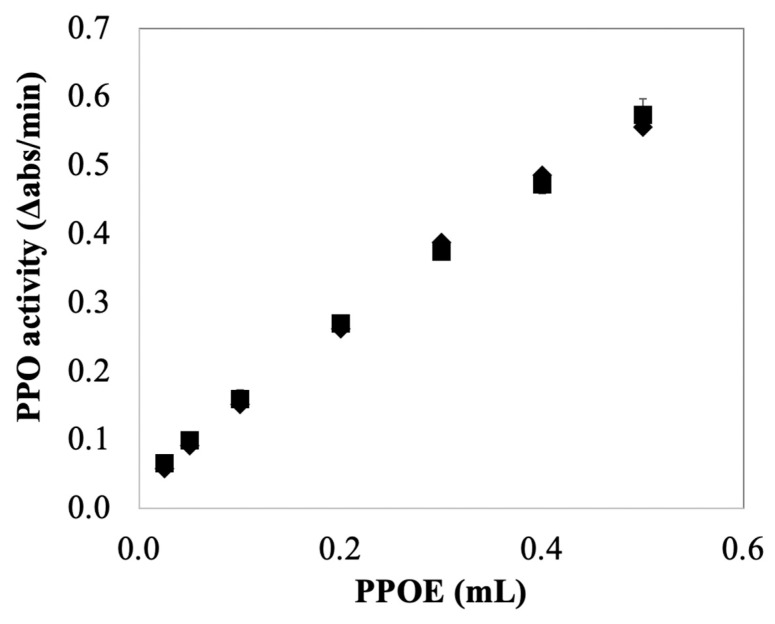
Effect of enzyme concentration, as an amount of PPO extract (PPOE) on PPO activity measurements using CA as a substrate. PPO reaction mixtures contained 1 mM CA and 50 mM sodium phosphate buffer, pH 7.0. Reactions were initiated by adding 0.2 mL PPOE with (■) and without (◆) AA (AA at 3 mM when present) to 2 mL reaction mixtures. Data points represent means ± standard deviations for triplicate assays. Data points without visible error bars have standard deviations smaller than the designated symbols.

## Data Availability

Data is contained within the article and Appendix A.

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
