# Peer review of "Role of Ascorbic Acid in the Extraction and Quantification of Potato Polyphenol Oxidase Activity"

_foods, 2021, doi:10.3390/foods10102486_

Round 1
Reviewer 1 Report
The authors have used potato extracts to determine the effect of ascorbic acid on polyphenol oxidation activity. Since ascorbic acid is used as a protective agent for oxidative enzymes, the findings of this study are important because they may provide an opportunity to review problems in conventional experimental systems. In addition, the results of this study are also important because plant extracts often contain ascorbic acid originally derived from the plant. However, the whole article is complicated and difficult to read and understand, and I felt that it needed a more clear context. In addition, I would like the authors to explain the following points.
1) In Line 277, what does the "se" indicate? I could not understand it.
2) I did not understand the theory in Line 338. You need to provide a citation here or in the sentence that leads to it. Or you should provide the explanation needed to derive this theory.
3) In Figure 3b, how does the absorbance change in the system without PPO? How does the absorbance change in the mixture of CA and AA? Depending on the results, the interpretation of this reaction may differ.
Reviewer 2 Report
The authors have a well written paper where they determined the effect of ascorbic acid on polyphenol oxidation activity using potato extracts. Ascorbic acid is usually used as a protective agent and so this study is of interest.
I have only two very minor comments here:
L97: please correct the citation here
L120: Please add „Solanum tuberosum” after Russet potato
Reviewer 3 Report
The article by Jiang and Penner investigated the role of AA in the extraction and quantification of potato PPO activity. Through carefully designed experiments, the authors have elucidated the different roles of AA as regrading to the protection of PPO activity in different systems/conditions. And in semi-purified system, the protecting effect is substrate-dependent. Overall, the article is well-written and the information presented is attracting to most food chemists. However, extracts from different species/foods would have different chemical environment, thus the findings of this study might be limited to potato extracts and hardly promoted to other circumstances.
Another minor point, the authors should revise the introduction section and make it more concise!
